# Scent of Health (S-O-H): Olfactory Multivariate Time Series Dataset for Non-Invasive Disease Screening

## Abstract

Exhaled breath analysis offers a promising, non-invasive alternative to traditional medical diagnostics. Electronic nose (eNose) sensors enable low-cost screening but progress is limited by small, site-specific datasets and sensor-specific temporal artifacts like baseline drift. We introduce Scent of Health (S-O-H), a large clinical eNose dataset with 1,027 patients across eight diagnostic groups, and reframe breath diagnosis as a realistic multivariate time series task. Our contribution includes curated temporal splits that control for sensor drift and mimic real-world deployment. We provide a reproducible benchmark with classical feature-based models, convolutional neural networks, and specialized time series classifiers. Our results demonstrate the dataset's utility, with methods achieving promising performance (e.g., ROC AUC up to 0.75 for lung cancer and 0.70 for hepatitis) while revealing significant gaps in robustness under drift and limited data. By releasing the dataset, splits, and code, we provide a foundational resource to advance research into robust, generalizable machine learning for clinical breathomics.

## 1 Introduction

Recent advances in artificial intelligence have been mainly driven by progress on richly annotated, high-volume datasets and architectures that can exploit temporal and high-dimensional structure (e.g., transformers and modern convolutional models). Yet, despite impressive successes in vision and language, specific sensory modalities remain underserved by openly available benchmarks and accompanying algorithmic studies. One of the most notable is the chemical sensing modalities underlying olfaction and breathomics. Exhaled breath contains complex mixtures of volatile organic compounds (VOCs) that encode clinically relevant metabolic information, and electronic-nose (eNose) technology offers a practical, portable route to digitize these signals for non-invasive diagnostics (Lee et al., 2024). However, the literature relies mainly on small cohorts specific to a single pathology, which limits the development of robust representation learning and reliable clinical evaluation for breath-based screening (Li et al., 2023). The scarcity of olfactory data stems from two primary challenges: the intrinsic difficulty of capturing odor information and the lack of standardized, affordable smell-digitization technology. While highly accurate, gold-standard methods like gas chromatography-mass spectrometry (GC-MS) are often prohibitively expensive and lack the usability for large-scale data collection. Consequently, researchers are seeking more cost-effective, accurate, and scalable sensor technologies, such as eNoses, to obtain reliable odor fingerprints. One promising direction is the on-chip printing of tailored metal oxide nanomaterials, a technique that enables the fabrication of dense arrays of analyte-specific microsensors. The resulting devices produce complex, data-rich response patterns to volatile compounds, creating a robust input for AI models in olfactory analytics (Goikhman et al., 2022; Gohel et al., 2024b).

However, two technical gaps hinder progress in machine learning for breathomics. First, the field lacks large, well-curated clinical collections that span multiple pathologies and support rigorous out-of-distribution and cross-site evaluation; this scarcity makes it difficult to study model generalization, sensor drift adaptation, and clinically meaningful performance thresholds. Recent efforts to assemble clinical breath molecule catalogs and to perform cross-site validation show promise but remain limited in scale or scope for broad benchmarking (Kuo et al., 2024). Second, eNose outputs are naturally multivariate time series with sensor-specific dynamics, cross-sensor correlations,

and measurement artifacts (e.g., baseline drift), so standard image/text architectures are not directly optimal without careful representation design and domain-aware augmentation.

In this work, we address both gaps. We introduce a large clinical eNose breathomics collection and cast odor diagnosis as a multivariate time series learning problem with realistic temporal and deployment challenges. Our dataset enables the development and evaluation of both classical and modern time series techniques, from strong feature-based tabular learners to dedicated time series architectures, and supports validation against temporal sensor drift and week-wise splits that mimic realistic deployment shifts. To establish informative baselines and highlight the algorithmic challenges that olfactory signals present, we evaluate a spectrum of approaches that have shown strong performance on time series tasks: random-kernel convolutional methods, deep convolutional ensembles, and recent self-supervised representation learning approaches for time series. These methods illustrate complementary trade-offs between speed, sample efficiency, and robustness to temporal perturbations; they also point to promising directions (data augmentation, pre-training, domain adaptation) for future work on sensor-based diagnostics.

The key contributions of this work are the following:

- **Scent of Health (S-O-H)**: A novel and extensive medical olfactory dataset comprising 1,027 patients across eight distinct groups (control and seven clinically significant disease groups), together with recommended train/test splits that control for temporal drift and realistic validation, it is the largest and most diverse dataset for this type of eNose device, capturing breath samples via a unique 17-sensor array of printed metal oxide (ZnO) microsensors on a temperature-controlled chip (Fig. 1).
- **Benchmarking and reproducible baselines**: We provide a comprehensive benchmark for odor classification as a multivariate time series analysis problem. It includes classical feature-based learners, and self-supervised representation baselines, evaluated under splits that expose drift and sample-size limitations.
- **Practical analysis of deployment challenges**: We quantify the effects of sensor drift and temporally concentrated sampling, and we report cross-validation strategies that minimize leakage while reflecting clinical deployment scenarios. These findings align with recent cross-site studies and underscore the need for domain adaptation in eNose applications.

## 2 RELATED WORK

Exhaled breath analysis, as a non-invasive method for diagnosis and disease monitoring, offers advantages over traditional methods like blood or urine analysis. Exhaled air contains VOCs, which are the end products of organic matter transformations in the body, and changes in the composition of VOCs can be used to diagnose diseases. In the last decade, this technology has been actively introduced into clinical practice as an alternative to traditional methods like GC-MS, which are labor-intensive, expensive, and lack portability (Chen et al., 2021).

eNose technology for breath analysis represents a rapidly advancing field with significant potential to transform medical diagnostics. Substantial progress has been made in demonstrating the clinical validity of this approach for various lung conditions, including COPD, asthma, and tuberculosis. The technology offers numerous advantages, including non-invasiveness, rapid results, cost-effectiveness, and potential for point-of-care testing. The eNose system is not inferior to this method and can detect mixtures of volatile metabolites even in low concentrations, without identifying individual chemicals. Machine learning methods allow the electronic nose to accurately identify odors using qualitative and quantitative analysis (Li et al., 2023).

Recently developed sensors (Goikhman et al., 2022) for the eNose applications allow for a large-scale study of the applicability of technology in diagnosing diseases by exhaled air. Prior work on the diagnosis of diseases by exhaled air focused on diseases of certain organ groups, for example, the digestive tract (Tiele et al., 2019), lungs and respiratory tract (Baldini et al., 2020), excretory system (Capuano et al., 2025), or used the data from a limited number of patients or did not use analog technologies of the eNose. Current eNose studies in healthcare typically also focus on individual diseases or anatomically-related conditions (Mortazavi et al., 2025) and rely on very limited datasets of tens to hundreds of patients, e.g., COPD (n=56, Acc. 82%) (Rodríguez-Aguilar et al., 2019), lung cancer (n=145, Spec. 84%) (Van de Goor et al., 2018), asthma (n=38, AUC 80%) (Tenero et al.,

Table 1: Comparison of S-O-H with Public Olfactory Datasets.

| Dataset | Modality | Task | Size | Clinical |
|---|---|---|---|---|
| Scent of Health (S-O-H) | eNose | Disease Screening | ∼1k patients | Yes |
| OlfactionBase (Sharma et al., 2022) | Molecular | Odor Perception | ∼5k molecules | No |
| M2OR (Lalis et al., 2024) | Molecular | Odor Perception | ∼50k molecules | No |
| SmellNet (Feng et al., 2025) | Molecular | Odor Perception | 50 substances | No |
| Olfaction-Vision-Language (France & Daescu, 2025) | Molecular + Vision | Cross-modal Learning | ∼5k samples | No |
| Multi-Labelled SMILES Odors (Lee et al., 2023) | Molecular | Odor/Property Prediction | ∼5k molecules | No |

2020) or tuberculosis (n=224, Spec. 87%) (Bruins et al., 2013). In most similar studies, the data are not disclosed, which hinders the advancement of olfactory modality for medical diagnostics.

To further situate our work within the broader ecosystem of available olfactory resources, we provide a comparative analysis in Table 1. As the table shows, existing public olfactory datasets focus predominantly on mapping chemical structures to perceptual odor labels, with no clinical focus. In contrast, the S-O-H dataset contributes to the emerging field of clinical breathomics by providing real-world sensor time series data from human patients for disease screening. This comparison highlights that S-O-H is, to the best of our knowledge, the first large-scale public resource to provide clinical-grade breath time series data, positioning it as a foundational benchmark for machine olfaction in healthcare rather than computational chemistry.

## 3 ENOSE HARDWARE

In this study, a multielectrode chip (Fig. 1) with 18 Pt (150 nm)/Ti (5 nm) strip co-planar electrodes is utilized to analyze the exhaled breath of patients. The chip, $10 \times 10$ mm$^2$, represents a silicon crystal with a silica layer of approximately 500 nm. Each pair of electrodes, distanced by 50 μm and with a functional material in between, forms an individual sensor segment, 17 in total. On-chip made two meander-shaped thermoresistors and two meander-shaped heaters enabled to control precisely the temperature of the chip surface during gas sensing measurements (Abayarathne et al., 2025; Gohel et al., 2024a;b). Subsequently, the prepared chip was wired to the ceramic card by ultrasonic bonding and installed in a gas-tight chamber with a chamber volume of 0.76 cm$^3$. The ceramic card with the chip was connected to a custom-made printed circuit board (PCB) to operate the sensor array and acquire the output signal at a sampling rate of 0.4 Hz. An IR pyrometer Kelvin Compact 1200D. was used to tune the temperature of the chip before and after the tests. Before testing the exhaled breath samples, the chip was kept at 300 ± 5 °C for 24 h in an air atmosphere for stabilization. The operational temperature of the multielectrode chip was maintained at 300 ± 5 °C.

The synthesis of functional materials for this study included the following route. A solution of lithium hydroxide ($LiOH$, 0.315 g, 0.075 mol) was added to 25 mL of absolute ethanol, using a dropping funnel. Afterwards, the obtained solution was added to solution of zinc nitrate ($Zn(NO_3)_2$), 1.49 g, 0.005 mol) and either indium/silver/cerium nitrate ($In(NO_3)_3/AgNO_3/Ce(NO_3)_4$) or nickel acetate ($Ni(CH_3COO)_2$, 0.00025 mol) in 25 mL of absolute ethanol. The addition was performed with vigorous stirring while the solution was cooled to 2 °C in an ice-water bath (Ge et al., 2017). The mixture was then stirred for 2 hours under the same conditions. Afterward, the precipitate was purified by centrifugation and rinsing alternately with ethanol and cyclohexane in 6 consecutive cycles. The precipitate was dried in dry air at 60 °C, then annealed in a furnace at 200 °C for 2 hours to finally get the corresponding powders. All chemicals were of at least analytical purity. The synthesized functional materials, i.e., zinc oxide or metal-doped zinc oxides ($ZnO, In-ZnO, Ag-ZnO, Ce-ZnO$, and $Ni-ZnO$) were placed

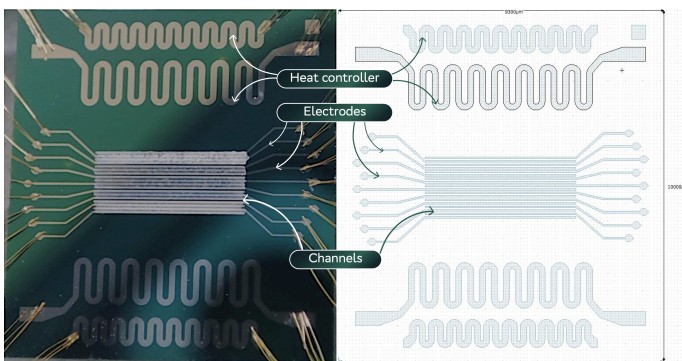

Figure 1: Our eNose device: physical implementation and schematic layout of the 17-element ZnO microsensor array.

on the top of the chip using a printing approach. The materials are deposited onto the chip surface using a REGEMAT 3D BIO V1 liquid bioprinter. As inks, particle suspensions are prepared with a particle mass ratio of 5 wt. % in an aqueous ethylene glycol solution (chemically pure, 60 wt. %). As a result, printed lines with an average width of ca. 300 µm were obtained, each covering three sensors. The prepared chip was annealed at 90 °C to remove the residual solvent.

# 4    DATA COLLECTION

## 4.1    PRE-COLLECTION PROCEDURES

The study cohort comprised patients with specific target nosologies and healthy volunteers. All participants provided written informed consent after being informed of the study's objectives and procedures. A physician-researcher screened each potential participant against predefined inclusion and exclusion criteria, and only eligible individuals were enrolled. The study protocol, including all ethical procedures, was approved by the Ethics Committee (see Section 8).

Upon enrollment, the physician completed a standardized participant questionnaire to record demographic and clinical data, including a unique study identification number, full name, year of birth, gender, clinical diagnosis coded with ICD-10 (World Health Organization, 2019), and corresponding Electronic Health Record (EHR) number.

Prior to breath sample collection, participants adhered to a standardized pre-sampling protocol designed to minimize confounding variables. This included a minimum 4-hour fasting period, abstinence from smoking for at least 4 hours, abstinence from alcohol for 48 hours, and the avoidance of perfumes and other strong odors on the day of sampling.

## 4.2    BREATH SAMPLE COLLECTION AND ANALYSIS

Exhaled breath samples were collected using individual, sterile, disposable 2-liter bags equipped with a check valve, chosen for their biocompatibility and standard medical-grade quality. A key constraint was that each participant could provide only a single sample for the entire study. The collection procedure was standardized and performed by trained laboratory staff. Participants first provided several exhalations to fill the bag, discarding the first part of each exhale to prioritize alveolar air.

For analysis, the sampling bag was connected to the eNose's intake port via a sealed valve. During sample analysis, an internal pumping system provided a constant, controlled airflow over the sensor array, mitigating variability. Each sampling bag was discarded after a single use, and the eNose's sensing chamber was purged with clean, dry air between samples to prevent cross-contamination. All collected samples were analyzed within 4 hours to ensure sample integrity.

Table 2: Distribution and demographic characteristics of the S-O-H dataset by diagnostic group.

| Diagnostic Group (ICD-10 Code) | N | % | Age, Mean | Age, Std | Male, % |
|---|---|---|---|---|---|
| Healthy Individuals (Z00) | 164 | 16.0 | 38.77 | 12.76 | 0.25 |
| Hepatitis B/C (B18) | 138 | 13.4 | 50.88 | 13.44 | 0.55 |
| Gastritis and Duodenitis (K29) | 138 | 13.4 | 52.32 | 16.27 | 0.36 |
| Non-alcoholic Fatty Liver Disease (K76) | 128 | 12.5 | 47.96 | 16.21 | 0.39 |
| Diabetes Mellitus Type II (E11) | 128 | 12.5 | 60.40 | 11.92 | 0.30 |
| Chronic Renal Failure (N18) | 128 | 12.5 | 59.82 | 14.61 | 0.50 |
| COPD (J44) | 100 | 9.8 | 64.16 | 10.43 | 0.64 |
| Malignant neoplasm of lungs (C34) | 100 | 9.8 | 66.71 | 8.63 | 0.73 |
| **Total** | **1027** | **100.0** | **52.9\*** | **15.1\*** | **0.45\*** |

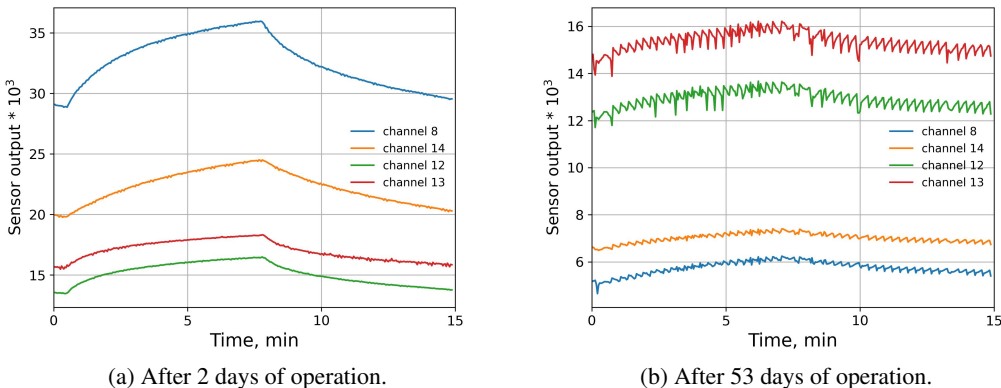

(a) After 2 days of operation.     (b) After 53 days of operation.

Figure 2: Progressive sensor signal deterioration.

### 4.3 COLLECTED DATA

Patient data is stored in a JSON structure. The primary sensor for olfactory analysis is the 'eNose', which records a multivariate time series of approximately 15 minutes in duration across 17 distinct channels. Auxiliary sensors within the device simultaneously monitor environmental parameters, including temperature, pressure, humidity, and $CO_2$ levels. The sensor output exhibits substantial variation in magnitude across its 17 channels, with each channel degrading at a unique rate over time (Fig. 2). The peak response for all channels occurs near the timestamp labeled 'endTimeGases' in the associated JSON file, while the 'durationSec' parameter records the total length of the measurement.

A total of 1027 samples were collected. The composition of the core patient cohort is detailed in Table 2. The minimum age of patients in the study is 18, and the maximum is 89. In general the distribution by gender was as follows: 567 (55.4%) women and 457 (44.6%) men.

### 4.4 VALIDATION SCHEME

The data was collected over a continuous 11-week period (Fig. 3). As exhaled air samples were gathered from outpatients, ensuring consistent collection for each disease became challenging. Additionally, eNose sensors are susceptible to degradation (Bosch et al., 2022), which may result in temporal sensor drift and lead to data leakage in disease classification. During the 71-day collection period, we observed a significant increase in signal fluctuations. Fig. 2 represents two waveforms from the same sensor at day 2 and day 53, respectively, illustrating increased fluctuations and changes in absolute signal levels.

To quantify the magnitude of this drift, we performed a dedicated analysis: we trained a CatBoost model to predict the day of measurement using only the raw sensor data. The model achieved

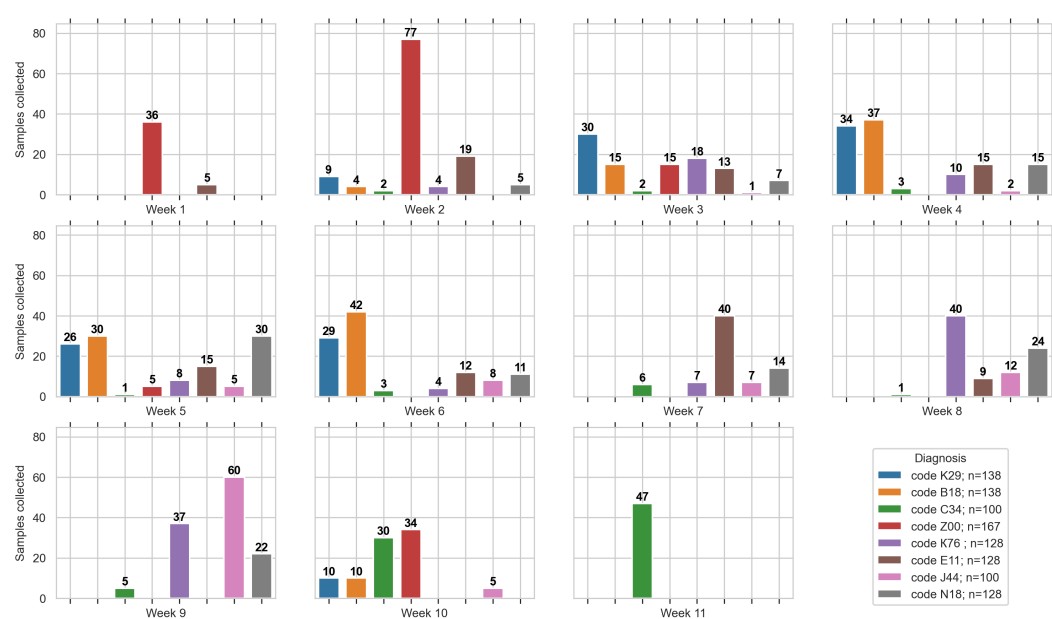

Figure 3: Distribution of the samples collected from patients throughout the study.

Table 3: Train (empty) / test (+) split by ICD-10 codes across weeks. Each medical condition uses different temporal segments for model evaluation.

| ICD-10 / Week | 1 | 2 | 3 | 4 | 5 | 6 | 7 | 8 | 9 | 10 | 11 |
|---|---|---|---|---|---|---|---|---|---|---|---|
| Z00 | | | | | | | | | + | + | + |
| E11 | | + | + | + | | | | | | | |
| K29 | | + | + | + | | | | | | | |
| K76 | | | + | + | + | | | | | | |
| B18 | | | + | + | | | | | | + | |
| C34 | | | | | | | | + | + | + | |
| N18 | | | + | + | + | | | | | + | |
| J44 | | | | | + | + | + | | | | + |

a median absolute error of 2-3 days, confirming that a strong temporal (non-stationary) signal is embedded in the recordings. Interestingly, this prediction relied primarily on shifts in the absolute baseline rather than on dynamic signal features, suggesting the disease-related kinetic information may be partially separable from the drift component.

To minimize the effect of the drift, we suggested an individual train/test split for each of the eight conditions, applying two criteria for selecting patients for validation. Firstly, the positive validation samples should be distant from the positive training samples. Secondly, the negative validation samples should include samples that are close in time to the negative train samples. We divided the experiments into weeks and used these chunks to create the train/test splits. Three of the weeks were used for validation, and the rest for training (Table 3).

While established correction techniques exist (Bosch et al., 2022; Liu et al., 2019; Tao et al., 2018), we deliberately abstained from applying them in our core benchmarks. Instead, we quantified the drift and preserved it within the proposed temporal splits. This design decision explicitly frames sensor drift robustness as a central, quantifiable challenge for the community. The S-O-H dataset, with its documented temporal stratification and auxiliary environmental logs, is positioned as a benchmark to develop and evaluate next-generation drift-robust algorithms, such as those based on domain adaptation or style transfer.

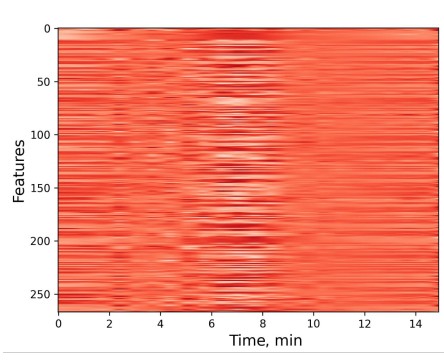

(a) Signal as image example.

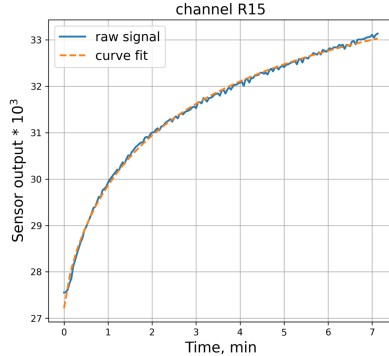

(b) Signal curve fitting example.

Figure 4: Time series representations.

## 5 BASELINE METHODS

### 5.1 CONVOLUTION-BASED METHODS

The key idea behind this approach is to treat time series data as images (Semenoglou et al., 2023; Hatami et al., 2018). To address sensor degradation, each time series is smoothed using a wavelet transformation and normalized via min-max scaling. Subsequently, polynomial features are extracted from the processed time series to capture non-linear dynamics between channels. These features are then combined with the original data to create a floating-point matrix that can be interpreted as an image. This result of the aggregation is illustrated in Fig. 4a. A neural network with several convolutional and fully connected layers processes the data and classifies it. The inference time for the whole process is less than one second on a laptop using only the CPU, so the algorithm could be implemented on embedded systems. Additionally, we applied pretrained ResNet18 (He et al., 2016) and finetuned it for binary classification providing the same floating-point matrix of polynomial features as input data. We trained both of these methods with 16 random splits of the training data, using 15% for validation, while the test data remained fixed (see Table 3).

### 5.2 FEATURE-BASED METHODS

We assume that data from the eNose sensors exhibit a curve pattern with a saturation plateau, which can be modeled as a kinetic curve. We used other basic statistical features to describe our time series data (Faouzi, 2022). Prior to preprocessing, the time series data is temporally clipped using experiment-defined 'startTimeGases' and 'endTimeGases' parameters. The signal is then smoothed with a median filter to counter sensor degradation and normalized via min-max scaling applied independently to each channel. Initially, we fit each time series with a function:

$$f(t) = R_0 + \frac{R_{\max} - R_0}{1 + (t_{50}/t)^k} \tag{1}$$

by using curve_fit method from scipy.optimize library (Fig. 4b). From the four resulting function parameters we selected two sets of features: 1) LogFit_2: $t_{50}$, $k$; and 2) LogFit_4: $R_{max}$, $R_0$, $t_{50}$, $k$. We also compared this approach to 5 basic statistical parameters (Stats_5): minimum, maximum, mean, median and standard deviation values of each time series.

The feature engineering methods were applied exclusively to eight specific channels from the eNose array, as these channels exhibited consistent signaling over time, a crucial requirement for effective feature extraction. The selection of these channels was determined *a priori* by following the chip manufacturer's recommendation, which was based on their extensive material science expertise and confidence in the consistent performance and chemical response of the printed substances on the channels. The resulting feature vectors for each patient comprised 32, 16, 24, and 40 elements, respectively, corresponding to each feature set. We adopted the following hyperparameters

for classification models: 1500 iterations, learning rate 0.05, max tree depth 10, L2 leaf regularization (l2_leaf_reg) 8 for CatBoost (Prokhorenkova et al., 2018); 5000 iterations, learning rate 0.07, max tree depth 8, L2 penalty (reg_lambda) 2 for XGBoost (Chen & Guestrin, 2016); 2000 iterations, learning rate 0.07, max tree depth 8, L2 regularization 0.2 for HistGradientBoost; 200 estimators and Gini criterion for RandomForest. We also trained each of these methods with 16 random train splits, using 15% for validation and early stopping.

### 5.3 GRAPH-BASED APPROACH

Given the eNose sensors' principle that channel-to-channel ratios are different between samples of different origin, we hypothesized that graph architecture could help leverage this distinct feature. We utilized Graphormer model (Ying et al., 2021) and trained it with our dataset for binary classification, following the graph representation outline described by Hu et al. (2020).

Our graph is a complete graph with 8 nodes, each one representing a single sensor channel. Node feature includes 9 parameters, derived from a corresponding time series: for a series $s_i$ a feature vector is defined as $x_i = \{\min(s), \max(s), \mathrm{mean}(s), \mathrm{std}(s), k, t_{50}, a_0, a_1, a_2\}$. For the definition of $k$ and $t_{50}$ values see function (Eq. 1). For additional $a_0$, $a_1$ and $a_2$ values we employed another approach which approximated the temporal signals using Chebyshev polynomials.

$$f(t) = a_0 + a_1 * T_1(t) - a_2 * T_2(t), \tag{2}$$

where $T_1$, $T_2$ denote the first- and second-order Chebyshev polynomials, respectively. Edge feature includes 3 parameters, derived from the two time series corresponding to the connected vertices. For a pair of series $s_i$ and $s_j$ feature vector is defined as $x_{ij} = \{\mathrm{abs}(\min(s_i) - \min(s_j)), \mathrm{abs}(\max(s_i) - \max(s_j)), \mathrm{std}(s_i)^2 / \mathrm{std}(s_j)^2\}$. Before any feature vector was calculated, sets of time series related to the same sample were normalized independently to preserve channel-to-channel ratios, while eliminating baseline-mediated drift. Each value was discretized into 32 bins and the resulting vectors contained of 9 and 3 integer values for a node and edge representations respectively.

### 5.4 TIME SERIES CLASSIFICATION

As the data collected from the eNose device constitutes a multivariate time series, our benchmarking would be incomplete without the training of advanced models for time series classification. To establish comprehensive baseline performance, we evaluated three methodologies: an LSTM model, the InceptionTime network (Ismail Fawaz et al., 2020), and the unsupervised TS2Vec framework (Yue et al., 2022). For our experiments, we implemented a 3-layer LSTM with a hidden state size of 128. The InceptionTimeClassifier was trained for 70 epochs with default hyperparameters. TS2Vec was trained with a learning rate of 0.001 and a batch size of 64 over 40 epochs. Subsequently, we employed logistic regression to classify the vector representations learned by TS2Vec. The standard deviation of the ROC AUC metric was estimated in the same manner as in our other experiments.

## 6 EXPERIMENTAL RESULTS

### 6.1 ODOR CLASSIFICATION

The ROC AUC of the methods described above applied to binary (condition is present or not) classification for 8 diseases from our benchmark are presented in Table 4. Here, first, feature-based classification with ensemble methods indicate two key insights for this straightforward approach. First, the inclusion of the minimum and maximum time series values does not significantly affect model performance, as the preprocessing steps render these parameters ineffective. For each ensemble method the difference between scores for LogFit_2 and LogFit_4 feature sets is negligible. Moreover, the fitted function parameters appear to be suboptimal and fail to fully capture the information contained in the time series data.

Second, among convolution-based methods, our relatively shallow CNN surpassed the fine-tuned ResNet, achieving the highest score in classifying healthy patients and demonstrating greater stability than ResNet. While the feature-based approach performs better in classifying fatty liver disease (K76) and hepatitis (B18), convolutional methods show promising results for malignant neoplasms of the lungs (C34), chronic renal failure (N18) and classifying healthy patients.

Table 4: Disease classification ROC AUC (mean ± std, **best values**) for our train/test split

| Model | Features | Z00 | E11 | K29 | K76 | B18 | C34 | N18 | J44 |
|---|---|---|---|---|---|---|---|---|---|
| CatBoost | LogFit_2 | 0.465 ±0.012 | 0.516 ±0.027 | 0.448 ±0.057 | 0.612 ±0.045 | 0.686 ±0.033 | 0.503 ±0.029 | 0.598 ±0.040 | 0.448 ±0.023 |
| | LogFit_4 | 0.468 ±0.006 | 0.478 ±0.020 | 0.425 ±0.033 | 0.600 ±0.061 | 0.677 ±0.028 | 0.503 ±0.028 | 0.615 ±0.028 | 0.452 ±0.018 |
| | Stats_5 | 0.507 ±0.016 | 0.551 ±0.028 | 0.453 ±0.032 | 0.648 ±0.027 | **0.698** ±0.011 | 0.530 ±0.012 | 0.541 ±0.021 | 0.435 ±0.025 |
| XGBoost | LogFit_2 | 0.455 ±0.040 | 0.508 ±0.034 | 0.458 ±0.028 | 0.545 ±0.061 | 0.605 ±0.057 | 0.549 ±0.061 | 0.507 ±0.025 | 0.462 ±0.027 |
| | LogFit_4 | 0.497 ±0.008 | 0.501 ±0.022 | 0.455 ±0.049 | 0.546 ±0.036 | 0.623 ±0.063 | 0.541 ±0.061 | 0.506 ±0.040 | 0.491 ±0.030 |
| | Stats_5 | 0.565 ±0.013 | 0.499 ±0.044 | 0.548 ±0.057 | 0.561 ±0.057 | 0.607 ±0.040 | 0.578 ±0.035 | 0.528 ±0.048 | 0.445 ±0.046 |
| HGBoost | LogFit_2 | 0.411 ±0.038 | 0.511 ±0.045 | 0.476 ±0.045 | 0.614 ±0.042 | 0.610 ±0.044 | 0.496 ±0.036 | 0.521 ±0.030 | 0.456 ±0.037 |
| | LogFit_4 | 0.440 ±0.042 | 0.495 ±0.031 | 0.470 ±0.036 | 0.563 ±0.043 | 0.676 ±0.034 | 0.463 ±0.032 | 0.515 ±0.029 | 0.481 ±0.033 |
| | Stats_5 | 0.510 ±0.029 | 0.526 ±0.029 | 0.515 ±0.060 | 0.615 ±0.057 | 0.651 ±0.030 | 0.580 ±0.022 | 0.561 ±0.045 | 0.429 ±0.027 |
| RandomForest | LogFit_2 | 0.479 ±0.041 | 0.491 ±0.044 | 0.471 ±0.060 | 0.588 ±0.045 | 0.627 ±0.029 | 0.519 ±0.043 | 0.530 ±0.032 | 0.475 ±0.043 |
| | LogFit_4 | 0.482 ±0.041 | 0.503 ±0.041 | 0.466 ±0.046 | 0.599 ±0.051 | 0.630 ±0.061 | 0.472 ±0.047 | 0.547 ±0.055 | 0.496 ±0.031 |
| | Stats_5 | 0.504 ±0.029 | 0.502 ±0.048 | 0.448 ±0.052 | 0.561 ±0.080 | 0.648 ±0.036 | 0.486 ±0.027 | 0.534 ±0.044 | 0.441 ±0.035 |
| CNN | Polynomial | **0.600** ±0.022 | 0.468 ±0.035 | 0.545 ±0.037 | 0.403 ±0.023 | 0.511 ±0.015 | 0.700 ±0.028 | 0.636 ±0.055 | 0.470 ±0.027 |
| ResNet18 | | 0.590 ±0.061 | 0.517 ±0.039 | 0.516 ±0.042 | 0.505 ±0.030 | 0.575 ±0.042 | 0.612 ±0.033 | 0.630 ±0.041 | 0.464 ±0.049 |
| Graphormer | Log&Poly | 0.578 ±0.118 | 0.554 ±0.075 | 0.513 ±0.051 | 0.502 ±0.113 | 0.462 ±0.044 | 0.630 ±0.059 | 0.517 ±0.095 | **0.767** ±0.061 |
| InceptionTime | | 0.496 ±0.121 | 0.492 ±0.053 | 0.495 ±0.074 | 0.408 ±0.048 | 0.548 ±0.060 | **0.746** ±0.030 | 0.608 ±0.065 | 0.461 ±0.057 |
| TS2Vec | Time series | 0.480 ±0.029 | 0.466 ±0.032 | 0.453 ±0.092 | 0.538 ±0.034 | 0.644 ±0.017 | 0.579 ±0.030 | 0.654 ±0.028 | 0.470 ±0.049 |
| LSTM | | 0.484 ±0.108 | 0.493 ±0.030 | 0.342 ±0.064 | **0.686** ±0.017 | 0.613 ±0.010 | 0.691 ±0.033 | **0.658** ±0.032 | 0.626 ±0.031 |

Third, Graphormer's results exhibit significantly greater variability compared to other methods. We believe this could be attributed to the model potentially overfitting to our relatively small dataset. Finally, among the three time series classification methods evaluated, the LSTM approach demonstrated the most consistent classification performance, achieving significant prediction scores for 5 out of 8 conditions. Notably, InceptionTime yielded the highest score for the classification of malignant neoplasms of the lungs (C34) cases.

These results form a comprehensive benchmark covering multiple diseases, highlight the significant potential for disease screening using a metal oxide eNose. The best classification results were obtained for hepatitis and malignant lung formations, with ROC AUC values surpassing 0.7. Thus, it is just a first step toward this challenging setting; achieving clinical-grade performance will require further development and validation.

## 6.2 DEMOGRAPHIC CONFOUNDERS

A critical consideration in medical ML is the influence of demographic variables, which can be powerful predictors (Rajkomar et al., 2018). In our research, we intentionally preserved the natural age and gender distributions of the diseases to reflect a realistic screening scenario. However, to rigorously quantify the influence of these demographics versus the breath signal, we trained the CatBoost models using only age, only sex, and their combination to establish a demographic-only performance baseline (Table 5, left). The results reveal a nuanced picture. As expected from clinical

Table 5: Disease classification with demographic features and across sex-/age-based cohorts, AUC.

| | Demographics-only models | | | CatBoost on cohorts | | |
|---|---|---|---|---|---|---|
| | **Age** | **Gender** | **Age+Gender** | **Men** | **Women** | **Age $> 55$** |
| Z00 | $0.967_{\pm0.004}$ | $0.570_{\pm0.000}$ | $0.935_{\pm0.012}$ | $0.597_{\pm0.014}$ | $0.382_{\pm0.016}$ | - |
| E11 | $0.627_{\pm0.011}$ | $0.520_{\pm0.000}$ | $0.651_{\pm0.008}$ | $0.474_{\pm0.047}$ | $0.459_{\pm0.032}$ | $0.520_{\pm0.040}$ |
| K29 | $0.494_{\pm0.020}$ | $0.499_{\pm0.000}$ | $0.575_{\pm0.025}$ | $0.380_{\pm0.062}$ | $0.422_{\pm0.046}$ | $0.451_{\pm0.052}$ |
| K76 | $0.473_{\pm0.031}$ | $0.497_{\pm0.000}$ | $0.447_{\pm0.025}$ | $0.484_{\pm0.055}$ | $0.653_{\pm0.035}$ | $0.567_{\pm0.056}$ |
| B18 | $0.555_{\pm0.012}$ | $0.607_{\pm0.000}$ | $0.583_{\pm0.012}$ | $0.552_{\pm0.046}$ | $0.763_{\pm0.039}$ | $0.684_{\pm0.045}$ |
| C34 | $0.678_{\pm0.009}$ | $0.607_{\pm0.000}$ | $0.729_{\pm0.007}$ | $0.477_{\pm0.055}$ | $0.477_{\pm0.052}$ | $0.730_{\pm0.021}$ |
| N18 | $0.617_{\pm0.011}$ | $0.500_{\pm0.078}$ | $0.632_{\pm0.013}$ | $0.562_{\pm0.049}$ | $0.653_{\pm0.028}$ | $0.549_{\pm0.041}$ |
| J44 | $0.555_{\pm0.017}$ | $0.616_{\pm0.000}$ | $0.651_{\pm0.008}$ | $0.462_{\pm0.022}$ | $0.615_{\pm0.038}$ | $0.464_{\pm0.027}$ |

knowledge (Sung et al., 2021; Buist et al., 2007), models based solely on age and gender are highly predictive for certain diseases like lung cancer (C34) and COPD (J44), confirming their strong demographic risk profiles. Furthermore, the high AUC ($>$0.9) for classifying the healthy cohort (Z00) against all diseases based on age alone is consistent with the known increase in morbidity with age.

Crucially, for other conditions such as liver disease (K76) and hepatitis (B18), the model using the full breath data significantly outperforms the demographics-only baseline. This corroborates that for these diseases, the model captures a strong, discriminative signal from breath VOCs that is independent of age or gender. The analysis confirms that while demographic information is inherently encoded in breath and is useful for screening, ML methods can capture additional, clinically relevant biological signals.

In addition, we split patients into cohorts – male vs. female and younger vs. older ($>$55 years, with 55 being the median age) – and computed ROC AUC for these cohorts (Table 5, right). Diseases K76, B18, and N18, which we successfully recognized using CatBoost on the full dataset, are also accurately recognized under this cohort-wise analysis, although the classification performance is poorer in the male cohort. Moreover, for K29, knowing gender and age does not allow reliable classification of the disease, whereas our methods make this possible based on eNose data analysis.

# 7 CONCLUSION AND FUTURE WORKS

This paper presents Scent of Health, a large clinical eNose breathomics collection with a reproducible benchmark and a suite of baselines that expose the practical challenges of olfactory time series, particularly sensor drift, limited labeled data, and cross-site variability. Our experiments show that while modern time series classifiers provide strong starting points, significant gaps remain in robustness and generalization under realistic temporal splits, highlighting the need for targeted augmentation, domain-adaptive pretraining, and sensor-aware modeling. By releasing the data, splits, and code, we aim to catalyze machine-learning research on olfactory data, from advanced time series architectures and pre-training strategies to robust adaptation techniques for sensor networks, and to accelerate the translation of eNose technology toward reliable, non-invasive disease screening. Future work will focus on scalable pre-training, principled drift correction, and prospective clinical validation to transition eNose systems from promising prototypes into reliable real-world tools.

While our benchmark demonstrates feasibility, it also outlines a clear path for future work to advance towards clinical utility. Key directions include expanding data collection to ensure temporal balance, enabling robust multi-fold validation, and incorporating a wider variety of sensor types. Furthermore, our baseline methods indicate that certain conditions like gastritis and diabetes may require specialized sensor configurations or novel architectures for effective classification, presenting a compelling challenge for the machine learning community. To directly support this future work, we are are going to evolve the S-O-H dataset into a growing, multi-center resource. An expansion phase is already underway, with 44 new tuberculosis samples collected at a second clinical site towards a target of 128. The initial 44 samples have already been added to the dataset, with the full cohort to be incorporated upon completion, ensuring S-O-H remains a vital and expanding benchmark for breathomics.

## 8 ETHICS STATEMENT

This study received ethics approval from the Local Ethics Committee at the ***Medical Research Organization*** (anonymized for peer review). Prior to their inclusion in the study, all participants provided written informed consent. The informed consent form as well as inclusion and exclusion criteria are available at `https://figshare.com/s/3b15245401fb5e1d8ef9`.

## 9 REPRODUCIBILITY STATEMENT

Once the paper is accepted, we will make our dataset publicly available and provide a DOI. For review purposes, the csv and json files are temporarily available at: `https://figshare.com/s/1db327052c2ffa570e64`. The code is available at `https://anonymous.4open.science/r/enose-3991/README.md`.

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
