# OpenReview forum: "Scent of Health (S-O-H): Olfactory Multivariate Time-Series Dataset for Non-Invasive Disease Screening"
_ICLR.cc/2026/Conference — Submitted to ICLR 2026_

### Official Review · Reviewer_8JQD · 2025-10-20

**Soundness:** 2
**Presentation:** 2
**Contribution:** 3
**Rating:** 6
**Confidence:** 3

**Summary:**

This paper introduces Scent of Health (S-O-H), an openly available olfactory multivariate time-series dataset for disease screening using electronic-nose (eNose) sensing.
It comprises 1,027 patient breath samples across eight diagnostic categories (healthy and seven disease conditions such as lung cancer, diabetes, and hepatic disorders), collected with a 17-sensor ZnO-based printed metal-oxide array. Each sensor produces 15-minute temporal recordings under controlled environmental conditions (temperature, humidity, CO₂, pressure).

The authors frame breath analysis as a multivariate time-series classification problem and benchmark:

CNN-based odor maps, treating sensor readings as spatiotemporal images, and

Feature-based ensemble models (e.g., CatBoost) using kinetic and statistical descriptors.

Temporal train/test splits emulate sensor drift and deployment chronology. CNNs achieve ROC-AUC ≈ 0.71 for some diseases, while feature-based models perform better for metabolic disorders.
All preprocessing scripts, splits, and metadata are planned for public release.

**Strengths:**

*High clinical and societal relevance:* Demonstrates non-invasive disease screening through breath sensing — an emerging, patient-friendly diagnostic approach.

*Open, reproducible resource:* One of the few publicly accessible, sensor-level olfactory datasets with detailed documentation and ethical clearance.

*Temporal realism:* Time-ordered splits realistically capture drift and device aging, a key challenge in olfaction research.

*Comprehensive methodology:* Clear sensor fabrication details, sampling setup, and preprocessing pipeline.

*Balanced baselines:* Includes both classical ML and deep architectures for fair benchmarking.

*Community orientation:* Code and data release encourage standardized evaluation in breathomics.

**Weaknesses:**

Moderate algorithmic novelty: CNN and CatBoost baselines are standard; no architecture tailored for drift-robust time-series olfaction is proposed.

*Limited diagnostic performance:* Best ROC-AUC ≈ 0.71 — below thresholds for practical screening.

*Single-center collection:* All samples originate from one institution, limiting generalization.

*Dataset imbalance:* Certain disease classes are underrepresented or temporally clustered.

*Drift unmodeled:* Drift is acknowledged but not explicitly quantified or corrected algorithmically.

*Missing broader dataset context:* The paper does not situate S-O-H relative to other open olfactory resources such as Olfaction-Vision-Language, SmellNet, OlfactionBase, M2OR, or Multi-Labelled SMILES Odors. A short comparison or feature-coverage table would clarify its unique positioning.

**Questions:**

This work signals a quiet but significant evolution — the maturation of machine olfaction from isolated prototypes toward shared, structured data ecosystems.
S-O-H transforms something intangible — the scent of human metabolism — into analyzable, time-bound data, bridging clinical sensing and AI reproducibility.

Its deeper value lies not in accuracy metrics but in infrastructure: it defines how the field can speak a common language. By quantifying sensor drift, linking breath profiles to diseases, and releasing data transparently, the authors lay groundwork for a research culture where olfactory AI becomes testable, comparable, and cumulative.

Yet, the study also exposes a limitation fundamental to the domain: olfactory signals are memory-laden. They encode environmental and temporal context as much as disease.
Recognizing that entanglement — and designing models resilient to it — will determine the future of digital smell analytics.

While algorithmic novelty is limited, the dataset’s openness, design realism, and reproducibility make it a meaningful benchmark contribution. To strengthen impact, the authors should:

1. Discuss overlap with existing open olfactory datasets.
2. Add metrics or models addressing sensor drift quantitatively.
3. Outline dataset maintenance and update strategy for long-term usability

I expect the authors to defend or rebut the points in the weakness section during the rebuttal phase. However, without that also this paper can be accepted as is as a Data Track / Benchmark Paper

**Update**:  After some internal discussion, I recognize that dataset-focused contributions should also be evaluated for strong methodological novelty. Accordingly, I have adjusted my scoring.

---

> ### Author Response · Authors · 2025-11-21
>
> Dear Reviewer,
>
> Thank you for your careful review of our work and valuable insights!
>
> Let us provide responses to your comments:
>
> **Moderate algorithmic novelty**
>
> The core contribution of our work is the dataset and its rigorous temporal evaluation, not a novel architecture. As shown in our responses to Reviewers hip3 (question **1**) and VA5c (weakness **ii**), we have benchmarked several SOTA time-series models (e.g., TS2Vec, InceptionTime). Our results indicate that the primary challenge lies in the data's inherent drift and variability, a core problem we frame for the community to solve.
>
> **Limited diagnostic performance**
>
> We fully agree that for now and for such a challenging task the performance is below clinical screening thresholds. We have addressed this point directly in our response to Reviewer hip3 (question **4**), reiterating that our work aims to provide a foundational benchmark to illustrate the field's challenges, not to claim clinical readiness.
>
> **Single-center collection**
>
> We acknowledge this limitation. As detailed in our response to Reviewer VA5c (weakness **ix**), we have an ongoing data collection at a second clinical site to address generalization, with 44 new samples already acquired. We will document this expansion in the revised paper.
>
> **Dataset imbalance**
>
> We acknowledge the class imbalance and temporal clustering. This is not an artifact of our collection but a reflection of epidemiological reality and patient flow at the clinical site. We deliberately preserved this distribution because it accurately mimics a realistic deployment scenario where a screening tool would encounter a naturally imbalanced population. While this poses a challenge for model training, it provides a more truthful benchmark for evaluating real-world performance.
>
> **Drift unmodeled**
>
> We agree that a deeper analysis of drift is crucial. While explicit algorithmic correction is a key challenge for future work, we have performed a new analysis to try to directly quantify the sensor drift.
>
> To measure the magnitude of the temporal drift, we trained a CatBoost model to predict the day of the study (over a 71-day collection period) using only the raw sensor data. The results demonstrate a clear temporal signal: the mean absolute error of this prediction was between 3 to 7 days, depending on the sensor channel. This confirms that a significant amount of temporally-variant information (the drift) is encoded in the sensor readings.
>
> Our analysis suggests that predicting the disease class relies on both the constant sensor baseline and the dynamic response (kinetic shape), whereas predicting the day is more dependent on the baseline. This indicates that the disease-related signal is partially separable from the drift, though they are quite entangled.
>
> We fully acknowledge that completely removing this drift is a profound challenge. The search for optimal correction methods would require repeated measurements of the same breath sample over time to obtain paired breath samples, which was not feasible in our current data collection setup. One possible option for the further development of the S-O-H dataset could be the paired collection of patient breath samples with the "preservation" of a second sample to obtain a drifted fingerprint for calibration.
>
> Given the current state of the S-O-H dataset, we have framed the drift problem explicitly as one of core challenges for the community. The dataset, with its clear temporal stratification, is designed to be the benchmark for developing such drift-robust algorithms. We believe that methods like domain adaptation, style-transfer, or the use of environmental auxiliary signals are promising future directions that can now be tested rigorously using S-O-H.

---

> > ### Author Response · Authors · 2025-11-21
> >
> > **Missing broader dataset context**
> >
> > We agree that situating S-O-H within the broader ecosystem of olfactory resources is important, and we will update the paper accordingly.
> >
> > We have created a comparative table to clarify its unique positioning:
> >
> > | Dataset | Primary Modality | Primary Task | Data Type | Size | Clinical Focus |
> > |---------|------------------|---------------|-----------|-----------------|----------------|
> > | Scent of Health (S-O-H) | eNose Sensor Array | Disease Screening | Multivariate Time-Series | ~1000 patients | Yes |
> > | OlfactionBase [*1*] | Molecular Structure | Odor Perception | SMILES & Labels | ~5000 molecules | No |
> > | M2OR [*2*] | Molecular Structure | Odor Perception | SMILES & Labels | ~50000 molecules | No |
> > | SmellNet [*3*] | Molecular Structure | Odor Perception | Molecular Graphs & Labels | 50 substances | No |
> > | Olfaction-Vision-Language [*4*] | Molecular Structure + Image | Cross-modal Odor Learning | SMILES, Labels, Images | ~5000 samples | No |
> > | Multi-Labelled SMILES Odors [*5*] | Molecular Structure | Odor/Property Prediction | SMILES, Labels | ~5000 molecules | No |
> >
> > As the table shows, other olfactory datasets focus on mapping chemical structures to perceptual odor labels with no clinical focus. In contrast, S-O-H contributes to the emerging field of clinical breathomics, providing real-world sensor time-series data from human patients for disease screening.
> >
> > This comparison highlights that S-O-H is, to the best of our knowledge, the first large-scale public resource to provide clinical-grade breath time-series data, positioning it as a foundational benchmark for machine olfaction in healthcare rather than computational chemistry.
> >
> > *1. Sharma, Anju, et al. "OlfactionBase: a repository to explore odors, odorants, olfactory receptors and odorant–receptor interactions." Nucleic Acids Research 50.D1 (2022): D678-D686.*
> >
> > *2. Lalis, Maxence, et al. "M2OR: a database of olfactory receptor–odorant pairs for understanding the molecular mechanisms of olfaction." Nucleic Acids Research 52.D1 (2024): D1370-D1379.*
> >
> > *3. Feng, Dewei, et al. "SMELLNET: A Large-scale Dataset for Real-world Smell Recognition." arXiv preprint arXiv:2506.00239 (2025).*
> >
> > *4. France, Kordel K., and Ovidiu Daescu. "Diffusion Graph Neural Networks and Dataset for Robust Olfactory Navigation in Hazard Robotics." arXiv [Cs.RO], 2025, arxiv.org/abs/2506.00455. arXiv.*
> >
> > *5. Lee, Brian K., et al. "A principal odor map unifies diverse tasks in olfactory perception." Science 381.6661 (2023): 999-1006.*

---

### Official Review · Reviewer_hip3 · 2025-10-30

**Soundness:** 3
**Presentation:** 2
**Contribution:** 2
**Rating:** 4
**Confidence:** 4

**Summary:**

The paper introduces Scent of Health (S-O-H) — a new multivariate time-series dataset for breath-based disease screening using electronic-nose (eNose) sensors. It represents one of the first large-scale efforts (1,027 participants across eight disease groups) to systematize olfactory sensing data and frame odor diagnosis as a time-series learning task. The authors provide baseline benchmarks using classical feature-based methods and CNNs, highlighting challenges such as sensor drift and temporal data splits.

**Strengths:**

1. New dataset contribution, the paper proposes olfactory or breathomics datasets limited exist for ML research.
2. The dataset covers over 1,000 participants with multiple diagnostic categories, enabling broader modeling and validation.
3. Provides baseline results that expose important challenges like drift, temporal bias, and limited labels.
4. Detailed sensor and protocol description.

**Weaknesses:**

1. Incomplete benchmarking scope: Evaluations use relatively simple CNN and CatBoost models but omit established or state-of-the-art time-series methods such as Transformers, InceptionTime, TS2Vec, Autoformer, Mamba, or masked-autoencoder approaches.

2. Lack of generalization studies: No exploration of cross-cohort, cross-time, or cross-sensor generalization, which is essential for deployment realism.

3. Limited statistical validation: The paper lacks multiple runs, variance reporting, or ablations to confirm robustness.

4. Clinical claims ahead of evidence: While non-invasive screening is promising, reported ROC-AUCs (~0.7) and limited sampling per subject make clinical readiness uncertain. More clinical valid evaluation metrics can be also reported.

**Questions:**

see weakness

---

> ### Author Response · Authors · 2025-11-21
>
> Dear Reviewer,
>
> Thank you for your helpful feedback on our work!
>
> Bellow our answers to the mentioned weaknesses:
>
> **1. Incomplete benchmarking scope**
>
> We have addressed a similar concern regarding model selection in our response to Reviewer VA5c (point **ii**). First of all, we kindly refer the reviewer to our answer above (https://openreview.net/forum?id=JTK6nljnag&noteId=OI2qDKRQEJ).
>
> Second, we agree that benchmarking against SOTA time-series methods is essential to fully characterize the challenges and opportunities within the S-O-H dataset.
>
> We have conducted further experiments with several SOTA methods as suggested. The results, using the same temporally-stratified train/test split to ensure a fair comparison, are presented below:
>
> | Method    | Z00           | E11           | K29           | K76           | B18           | C34           | N18           | J44           |
> |-----------|---------------|---------------|---------------|---------------|---------------|---------------|---------------|---------------|
> | InceptionTime   | 0.496 ± 0.121 | 0.492 ± 0.053 | 0.495 ± 0.074 | 0.408 ± 0.048 | 0.548 ± 0.060 | 0.746 ± 0.030 | 0.608 ± 0.065 | 0.461 ± 0.057 |
> | TS2Vec  | 0.480 ± 0.029 | 0.466 ± 0.032 | 0.453 ± 0.092 | 0.538 ± 0.034 | 0.644 ± 0.017 | 0.579 ± 0.030 | 0.654 ± 0.028 | 0.470 ± 0.049 |
> | LSTM   | 0.484 ± 0.108 | 0.493 ± 0.030 | 0.342 ± 0.064 | 0.686 ± 0.017 | 0.613 ± 0.010 | 0.691± 0.033 | 0.658 ± 0.032 | 0.626 ± 0.031 |
> | Graphormer [*1*]  | 0.578 ± 0.118 | 0.554 ± 0.075 | 0.513 ± 0.051 | 0.502 ± 0.113 | 0.462 ± 0.044 | 0.630 ± 0.059 | 0.517 ± 0.095 | 0.767 ± 0.061 |
>
> The results demonstrate that the proposed, relatively simple CNN and CatBoost baselines are highly competitive, often outperforming more complex SOTA time-series models on this specific dataset and evaluation protocol. This is a significant finding, as it suggests that the primary challenge in clinical breathomics may not be model complexity per se, but rather the fundamental issues of signal variability, drift, and dataset size that our paper highlights.
>
>
> *1. Ying, Chengxuan, et al. "Do transformers really perform badly for graph representation?." Advances in neural information processing systems 34 (2021): 28877-28888.*
>
>
> **2. Lack of generalization studies**
>
> We fully agree that understanding model performance across different subpopulations is essential for assessing real-world deployment potential. As mentioned in the answer to question **ix**) to Reviewer VA5c, we have initiated such data collection.
>
> Besides, we can immediately address cross-cohort generalization. We have performed this vital analysis within the S-O-H dataset to evaluate fairness and robustness across key demographic splits. Specifically, we evaluated one of our model on the held-out test set, segmenting the samples by sex and age (using the median age of 55 as a threshold). The results are as follows:
>
> | Disease Code | Overall | Male | Female | Young (≤55) | Old (>55) |
> |--------------|---------|------|--------|-------------|-----------|
> | Z00      | 0.462   | 0.593 | 0.372  | 0.462       | n/a         |
> | E11      | 0.486   | 0.553 | 0.433  | 0.452       | 0.533     |
> | K29      | 0.326   | 0.359 | 0.436  | 0.388       | 0.392     |
> | K76      | 0.618   | 0.556 | 0.630  | 0.712       | 0.568     |
> | B18      | 0.704   | 0.556 | 0.787  | 0.689       | 0.734     |
> | C34      | 0.515   | 0.443 | 0.454  | 0.275       | 0.724     |
> | N18      | 0.614   | 0.551 | 0.649  | 0.546       | 0.574     |
> | J44      | 0.480   | 0.464 | 0.638  | 0.370       | 0.438     |
>
> This analysis identifies specific cohorts where model performance generalizes well and where it does not. For instance, the model for liver disease (K76) shows a noticeable performance disparity, working significantly better for younger patients compared to older ones. These findings reveal specific disease-cohort pairs that require future mitigation strategies, such as cohort-specific model calibration or additional targeted data collection.

---

> ### Author Response · Authors · 2025-11-21
>
> **3. Limited statistical validation**
>
> Good point! As already shown in other replies with additional results, we update the metric values with std by varying the training data set through random sampling.
> Here we provide statistical validation for methods from our manuscript, we will carry out validation experiments for every baseline that we will add to our revised paper.
>
> *CNN (ours) with polynomial features.*
>
> | Z00           | E11           | K29           | K76           | B18           | C34           | N18           | J44           |
> |---------------|---------------|---------------|---------------|---------------|---------------|---------------|---------------|
> | 0.600 ± 0.022 | 0.468 ± 0.035 | 0.545 ± 0.037 | 0.403 ± 0.023 | 0.511 ± 0.015 | 0.700 ± 0.028 | 0.636 ± 0.055 | 0.470 ± 0.027 |
>
> *Catboost*
>
> | Method   | Z00           | E11           | K29           | K76           | B18           | C34           | N18           | J44           |
> |----------|---------------|---------------|---------------|---------------|---------------|---------------|---------------|---------------|
> | logfit_2 | 0.465 ± 0.012 | 0.516 ± 0.027 | 0.448 ± 0.057 | 0.612 ± 0.045 | 0.686 ± 0.033 | 0.503 ± 0.029 | 0.598 ± 0.040 | 0.448 ± 0.023 |
> | logfit_4 | 0.468 ± 0.006 | 0.478 ± 0.020 | 0.425 ± 0.033 | 0.600 ± 0.061 | 0.677 ± 0.028 | 0.503 ± 0.028 | 0.615 ± 0.028 | 0.452 ± 0.018 |
> | stats_3  | 0.507 ± 0.016 | 0.551 ± 0.028 | 0.453 ± 0.032 | 0.648 ± 0.027 | 0.698 ± 0.011 | 0.530 ± 0.012 | 0.541 ± 0.021 | 0.435 ± 0.025 |
> | stats_5  | 0.507 ± 0.016 | 0.551 ± 0.028 | 0.453 ± 0.032 | 0.648 ± 0.027 | 0.698 ± 0.011 | 0.530 ± 0.012 | 0.541 ± 0.021 | 0.435 ± 0.025 |
>
> **4. Clinical claims ahead of evidence**
>
> We thank the reviewer for this perspective, which aligns with our own view. We reiterate that the primary contribution of this work is the establishment of the S-O-H dataset as a foundational benchmark for the ML community. The reported performance (with ROC-AUC in 0.7 range) is intended not as a claim of clinical readiness, but as a baseline to illustrate the current challenges and opportunities for the field. We fully agree that a comprehensive clinical evaluation, including metrics like sensitivity and specificity at decision thresholds, is an essential next step that builds upon this foundational data resource.

---

### Official Review · Reviewer_M8f6 · 2025-11-01

**Soundness:** 3
**Presentation:** 2
**Contribution:** 3
**Rating:** 6
**Confidence:** 3

**Summary:**

Scent of Health (S-O-H) fills a notable gap in olfactory/breathomics ML with a sizable, carefully described, drift-aware collection and executable baselines; code/data availability (even if contingent) and a realistic temporal-split protocol raise its community value. However, the authors should should strengthen the quantitative evaluation (CIs, PR-AUC, calibration), analyze demographic/environmental confounds, report the missing self-supervised baselines, and document feature/channel selection strictly within training folds.

**Strengths:**

The strengths of the paper can be summarised as follows,

- Substantial, clinician-relevant dataset + clear problem framing. 1,027 participants across eight diagnostic groups (healthy + seven ICD-10 disease cohorts) measured on a printed metal-oxide eNose array with auxiliary environmental sensors. Single sample per participant helps avoid subject-level leakage.
- Drift-aware, temporally stratified evaluation. Collection spans 11 weeks; the paper proposes per-disease week-wise train/test splits to reduce leakage from sensor drift and temporally concentrated sampling, which is an important and realistic protocol choice for deployment settings.
- Hardware/system description supports reproducibility. Sensor fabrication, electrode layout, thermal control, sampling rate, storage schema and bagging protocol are explained with enough specificity to replicate data acquisition.
- Baseline breadth across modeling styles. Two families of baselines one with CNN treating the series as images after smoothing/normalization/feature aggregation; while the second is tabular CatBoost with kinetic-curve fits and basic statistics.

**Weaknesses:**

The weaknesses of the paper can be summarised as follows,

- Demographic confounding risk not analyzed. Disease cohorts differ markedly in age/sex (e.g., lung cancer and COPD groups older and more male than healthy). Without demographic balancing/adjustment or reporting of models with/without these covariates, results may reflect confounds rather than VOC signature.
- Potential internal inconsistencies / incomplete baselines. The text references contrastive/self-supervised baselines, but the main results table shows only CNN and CatBoost; claims such as "convolutional contrastive learning shows promising results" are not substantiated numerically in the table. Clarify or add the missing results.
- Choice of features/channels may induce selection bias. CatBoost uses only eight "stable" channels; criteria for selecting them (and whether selection used only training data per split) are not detailed, risking information peeking.
- Limited clinical utility metrics. Beyond ROC-AUCs, there’s no reporting of sensitivity at clinically relevant specificity (or vice-versa), decision curves, or subgroup performance (e.g., smokers vs non-smokers if logged), which are essential for a screening narrative.
- No cross-site validation; single-center collection. The dataset appears to be collected at one site/device configuration; claims of generalization under drift would be stronger with cross-site tests or at least leave-block-of-weeks-out CV with CIs.

**Questions:**

The questions for the authors as are posted below,

- Confound analysis. How do results change after adjusting for age/sex (e.g., propensity matching, adding them as nuisance covariates, or stratified evaluation)? Please report AUC/PR-AUC with and without demographic adjustment.
- Missing baselines. The text mentions contrastive/self-supervised methods; can you include their numeric results in the main table (with seeds/CIs) and specify training details?
- Temporal split robustness. Beyond the recommended three validation weeks, can you report leave-one-week-out CV (or multiple week-folds) with mean 95% CI to quantify variability under drift?
- Channel selection protocol. The CatBoost models use eight "stable" channels. How were they chosen (per-split on train only vs. globally)? Please document the selection rule and add a sensitivity analysis over channel subsets.
- Pre-processing clarity. Is the "weightlet" transform a typo for wavelet? I assume so but want to double check...
- Bagged-air protocol variability. Do you record breath volume/flow, or bag fill level? If not, could variable dilution explain some across-week variability? Any normalization by peak flow or a proxy?

---

> ### Author Response · Authors · 2025-11-19
>
> Dear Reviewer,
>
> We appreciate your time and valuable suggestions for improving our paper!
> Let us answer your questions:
>
> **Confound analysis.**
>
> This question is a continuation of the first concern. We agree that demographic confounders must be rigorously addressed. It is well-established in medical ML that patient age is often the most predictive feature in clinical models [*1*]. Our original approach - training on the entire dataset - reflects a real-world screening scenario where a model will encounter patients of all ages and sexes. We intentionally did not balance demographics to preserve this clinical realism, as diseases like type 2 diabetes, COPD, and lung cancer naturally have strong age and sex associations [*2, 3, 4*].
>
> Age and sex are not merely confounding variables; they are fundamental biological parameters that directly shape an individual's metabolic state and volatile organic compound (VOC) profile. Therefore, these demographic factors are implicitly encoded within the breath sample itself. Isolating the influence of age is always a challenging task.
>
> However, to directly answer the question and quantify the influence of demographics versus the breath signal, we conducted a new analysis. We retrained our CatBoost model under additional conditions, including models using only age, only gender, or both.
>
> The results (ROC-AUC) are as follows:
>
> | Disease          | log_fit4 | Age Only | Gender Only | Age+Gender |
> |------------------|-------------|----------|-------------|------------|
> | Z00    | 0.462       | 0.968    | 0.57        | 0.936      |
> | E11   | 0.486       | 0.63     | 0.52        | 0.66       |
> | K29  | 0.326       | 0.473    | 0.499       | 0.588      |
> | **K76**        | 0.618       | 0.466    | 0.497       | 0.438      |
> | **B18**  | 0.704       | 0.567    | 0.607       | 0.584      |
> | C34  | 0.515       | 0.676    | 0.607       | 0.74       |
> | N18         | 0.614       | 0.595    | 0.422       | 0.639      |
> | J44        | 0.48        | 0.556    | 0.616       | 0.649      |
>
> This analysis reveals a nuanced picture. For some diseases like lung cancer (C34) and COPD (J44), demographic models are highly predictive, confirming the known risk profiles. However, for other conditions like liver disease (K76) and hepatitis (B18), our model significantly outperforms the demographics only models. This provides clear evidence that for these specific diseases, the model is capturing a strong signal from the breath VOCs that is not a reflection of age or sex.
> We will include this analysis in the revised paper.
>
>
> *1. Rajkomar, Alvin, et al. "Scalable and accurate deep learning with electronic health records." NPJ digital medicine 1.1 (2018): 18.*
>
> *2. Dedov, I. I., Shestakova, M. V., & Galstyan, G. R. (2016). The prevalence of type 2 diabetes mellitus in the adult population of Russia (NATION study). Diabetes mellitus, 19(2), 104-112.*
>
> *3. Buist, A. Sonia, et al. "International variation in the prevalence of COPD (the BOLD Study): a population-based prevalence study." The Lancet 370.9589 (2007): 741-750.*
>
> *4. Sung, Hyuna, et al. "Global cancer statistics 2020: GLOBOCAN estimates of incidence and mortality worldwide for 36 cancers in 185 countries." CA: a cancer journal for clinicians 71.3 (2021): 209-249.*
>
>
> **Missing baselines.**
>
> This point was also raised by other reviewers. We have provided a response regarding the other baseline results in our answer to Reviewer VA5c (https://openreview.net/forum?id=JTK6nljnag&noteId=OI2qDKRQEJ). As discussed there, we will ensure these results are included in the main table of the revised paper and will expand the experiment section to provide the requested training details.
>
> **claims such as "convolutional contrastive learning...**
>
> We thank the reviewer for their careful reading. The mention of 'convolutional contrastive learning' was an error from a previous draft, this direction was not developed in our current work. We will remove this statement from the final manuscript to avoid confusion.

---

> > ### Author Response · Authors · 2025-11-19
> >
> > **Temporal split robustness.**
> >
> > We fully agree that a leave-one-week-out cross-validation scheme would be the ideal method to quantify performance variability under drift. However, this is unfortunately not feasible with the current dataset due to a fundamental constraint of clinical data collection.
> >
> > The core issue is that for any specific disease, the positive cases are not uniformly distributed across all weeks. As can be seen in Figure 2, most weeks contain positive samples for only a subset of diseases. Performing leave-one-week-out validation for a single disease would result in most test folds having zero positive instances, making it impossible to compute a meaningful ROC-AUC.
> >
> > Our proposed temporal split (training on weeks 1-3, testing on week 10) was carefully designed as the most robust alternative to this problem. It strictly prevents temporal data leakage and provides a realistic simulation of model deployment, where the model is trained on past data and evaluated on future data from a different temporal regime.
> >
> > To illustrate the critical importance of this strict split, we conducted an ad-hoc experiment using a temporally ill_split: training on week 2 and testing on week 1. The results for the healthy class (Z00) are demonstrative:
> >
> > | Disease          | temporal_split | ill_split |
> > |------------------|-------------|----------|
> > | Z00    | 0.468 ± 0.008       | 0.892 ± 0.007    |
> >
> > This artificially inflated ROC-AUC of 0.892 is a clear indicator of model overfitting to the specific sensor state and environmental conditions of a single week, rather than learning the genuine physiological signal. It starkly demonstrates how temporally incorrect splits can lead to severe data leakage and overly optimistic results.
> >
> > We acknowledge that a perfectly balanced temporal distribution of classes was not logistically possible under the strict protocols of our clinical study. Therefore, while we cannot provide the requested leave-one-week-out CV, we contend that our chosen split strategy is not only more rigorous for this dataset but also provides a more realistic and challenging benchmark for evaluating model robustness against temporal drift.
> >
> > **Channel selection protocol.**
> >
> > Thanks, this is an important question. The selection of the "stable" channels was determined *a priori* by following the chip manufacturer's recommendation. The manufacturer selected these specific sensor channels based on their extensive material science expertise and confidence in the consistent performance and chemical response of the printed substances on these tracks.
> > Therefore, the channel set is fixed and was applied globally before any model training, ensuring no data leakage from our clinical dataset. This represents a real-world scenario where a device is used with its predefined, optimal sensor configuration.
> >
> > **Pre-processing clarity.**
> >
> > Yes, that is correct! It is a typo and should be "wavelet". Thank you for catching it! The details of the transformation can be found in cell [14] of the *iclr_cnn.ipynb* notebook.
> >
> > **Bagged-air protocol variability.**
> >
> > Thank you for this important question regarding protocol standardization.
> >
> > To ensure sample consistency, we employed the following measures:
> >
> > 1. The use of sterile, 2-liter medical-grade bags equipped with a check valve.
> >
> > 2. Laboratory tech staff were trained to follow a strict protocol. Patients provided several exhalations to completely fill the bag, with the first part of each exhale discarded to prioritize alveolar air.
> >
> > 3. During sample analysis, an internal pumping system provided a constant, controlled airflow over the sensor array, mitigating variability.
> >
> > We acknowledge that minor variability in bag fill level is possible. Based on the bag's total volume and the collection procedure, we empirically estimate this potential dilution error to be ≤5% (approximately 100 ml). We did not record peak flow or use it for normalization in this study, but we agree this is a valuable consideration for future work.
> >
> >
> > **Limited clinical utility metrics.**
> >
> > This is a valuable suggestion. The primary focus of this work is to establish the S-O-H dataset as a foundational ML benchmark, which is why we prioritized standard ML evaluation metrics like ROC-AUC. While a full clinical validation is beyond the scope and format of this paper, we agree that metrics like sensitivity at fixed specificities are crucial for clinical translation.
> >
> > **No cross-site validation; single-center collection.**
> > We have provided a detailed response to the identical concern raised by Reviewer VA5c (Weakness **ix**). We kindly refer the reviewer to that response - https://openreview.net/forum?id=JTK6nljnag&noteId=ydv1sjRYjY.

---

### Official Review · Reviewer_VA5c · 2025-11-01

**Soundness:** 3
**Presentation:** 2
**Contribution:** 2
**Rating:** 4
**Confidence:** 5

**Summary:**

This paper introduces the Scent of Health dataset, a large-scale clinical breathomics collection designed to advance machine learning for eNose based disease diagnostics. The dataset comprises 1,027 patients across eight groups (one control and seven disease) and captures breath samples as multivariate time series (~15-minute) from a custom 17-sensor metal oxide microsensors along with an auxiliary sensor (that captures environmental parameters like temperature, pressure, humidity, and CO2 levels). The authors highlight two main challenges in the field: the scarcity of large, public clinical datasets and the difficulty of modeling eNose data, which is affected by sensor drift and temporal artifacts.

To address these, they provide

i) the dataset with recommended temporally stratified, week-wise train/test splits that account for sensor drift and propose a benchmark evaluating both feature-based (CatBoost) and deep learning (CNN) methods that reinterpret time series as images.

ii) The results demonstrate the potential of eNose technology for non-invasive screening while revealing significant challenges in robustness and generalization, paving the way for future work in domain adaptation and specialized time-series modeling. The paper provides code and temporary data links for reproducibility.

**Strengths:**

The authors have touched each dimension of originality, quality, clarity, and significance.

Originality: The primary contribution is the creation and public release of the S-O-H dataset. It is positioned as the largest and most diverse of its kind for this specific eNose technology, filling a critical gap in the availability of large, clinically annotated olfactory time-series data. The paper thoughtfully frames the problem around realistic deployment challenges, specifically temporal sensor drift and week-wise data distribution. The creation of temporal splits to prevent data leakage and better simulate real-world performance is a nuanced and highly relevant contribution beyond standard random splits.

Quality: The data collection protocol is described in detail, covering participant preparation, sensor chip design and fabrication (including material synthesis), and data logging procedures. This thoroughness inspires confidence in the dataset's integrity and supports reproducibility. The validation scheme is carefully motivated to reduce leakage due to temporally concentrated sampling. Also, for reproducibility, open-sourcing the dataset and code is a significant strength that will catalyze research in the community.

Clarity: The paper is well-written and logically structured. The motivation is clear, the technical gaps are well-articulated, architecture choices, validation strategies, and the contributions are succinctly stated. Figures like the sample distribution over weeks (Fig. 2) effectively communicate the core challenge of temporal concentration.

Significance: Given the increasing interest in non-invasive diagnostics, the dataset and this work provide a critical foundation for advancing AI in breathomics. The open release and reproducible baselines maximize potential impact across both medical and machine learning communities

**Weaknesses:**

i) At line 77, the authors mention using a unique 17-sensor array of printed ZnO-based metal-oxide microsensors on a temperature-controlled chip. However, the paper does not include a schematic of the sensor layout. Providing even a simplified diagram would greatly enhance clarity and help readers understand the sensing configuration and spatial arrangement of the sensors.

ii) The paper evaluates only two models, omitting other competitive baselines such as Random Forest, HistGradientBoost, XGBoost, or deep architectures like ResNet that can process time-series data as images. Including or at least discussing these alternatives would provide a more comprehensive and fair performance comparison.

iii) The choice of a generic CNN that treats the data as an image may not be the most effective or natural way to model the underlying temporal dynamics and cross-sensor correlations.

iv) Why were more established architectures like LSTM, InceptionTime, HIVE-COTE, or transformers not included as baselines? Was the relatively simple CNN structure chosen primarily for its speed and suitability for embedded deployment, and if so, could this be stated more explicitly?

v) While the temporal split is a major strength, the paper does not provide a detailed analysis of the drift. Quantifying the drift (e.g., using dimensionality reduction to visualize feature distribution shift across weeks) or directly evaluating baseline drift-correction methods would have strengthened the practical analysis of deployment challenges contribution.

vi) The authors correctly note the inability to perform multi-fold validation due to the temporal concentration of data. However, this limits the statistical robustness of the reported AUC scores. The results should be interpreted as initial findings rather than definitive performance metrics.

vii) The work motivates cross-site issues but presents data from a single device setting; even a small leave-session or leave-batch analysis or synthetic domain shift (temperature or humidity perturbations) would add evidence for external validity.

viii) The code is not visible at the given anonymous link. It shows the message "The requested file is not found".

ix) Are there plans for additional data collection to better balance the disease splits and enable more powerful multi-fold validation?

x) How does the sensor drift observed here compare quantitatively to other sensor deployments or to data in existing olfactory benchmarks?

xi) Was any analysis performed to ensure the models are learning from the breath signal and not from spurious correlations with, for example, the average age difference between healthy and disease groups?

**Questions:**

I would request authors to answer all points that are raised in the Weaknesses.

---

> ### Author Response · Authors · 2025-11-18
>
> Dear Reviewer,
>
> We sincerely appreciate your thoughtful feedback. Let us address each mentioned weakness below:
>
> **i)** Agreed. We will add this (https://postimg.cc/FkKHLgh9) side-by-side macro photo of the physical chip and its layout schematic.
>
> **iii)** This is a valid point. While a generic CNN may not be the most natural fit, we aimed to explore the efficacy of treating the data as an "odor image" to tap into the extensive toolkit of computer vision, similar to the success of spectrograms in audio processing. We agree that specialized time-series models are crucial for comparison, and as mentioned, we will be adding these baselines.
>
> **iv)** This is a valuable question. Our initial baseline selection was indeed guided by two primary factors, which we will state more explicitly in the revision. First, as the reviewer correctly guessed, the ultimate goal is a portable, low-cost medical diagnostic device. This makes computational efficiency and inference speed critical, favoring simpler architectures like our CNN. Second, we considered data scalability; many advanced architectures like Transformers require vast amounts of data for an effective pretraining, which exceeds our current dataset size.
> We agree that exploring these more established architectures is a crucial scientific direction, and as per previous comments, we will try to include several of them (e.g., LSTM, InceptionTime) in our expanded benchmarks.
>
> **v)** We agree that a deeper analysis of the drift would significantly strengthen the paper. We have begun preliminary work in this direction and, based on your feedback, will prioritize completing this analysis and incorporating it into the revised manuscript to better quantify the deployment challenge.
>
> **vi)** We agree with this observation. We will clarify in the paper that the results are initial benchmarks for this challenging setting, not definitive performance metrics.
>
> **vii-xi)** We thank the reviewer for this insightful point, which strongly echoes with the last weakness. Both concerns are fundamentally about isolating the true medical diagnostic signal of the breath from confounding factors and assessing model generalization. To address these issues, we will perform additional analyses along this line and will update you with the results by the end of November. Currently, we are considering a synthetic domain shift approach, applying artificial perturbations to the data to mimic changes that would occur in a different environment.
>
> **viii)** The correct and functional URL is: https://anonymous.4open.science/r/enos-8FD9
> We have verified it is accessible and will double-check that this link is correctly embedded in the final version of the paper.
>
> **ix)** Yes, we are. To address this exact point, we have an ongoing data collection effort for tuberculosis (a disease of high social concern) at a second clinical site using the same device. As of mid-November, we have collected 44 new samples, with a target of 128. We will update the public dataset with these new samples, even as the tuberculosis class is still incomplete, and include a note on this expansion in the revised paper.
>
> **x)** This is a valuable point for putting our dataset into context. A direct quantitative comparison to other deployments is challenging due to the scarcity of public clinical eNose datasets that document long-term drift. As established in the electronic nose literature [*1*], the standard practice of short-term data collection with random partitioning artificially boosts performance metrics, as models are evaluated on data from the same temporal domain. To address this, our work adopts a rigorous temporal splitting protocol that simulates real-world deployment conditions, thereby providing more realistic and reliable performance metrics. Qualitatively, the performance degradation we observe under this temporal protocol is significant and consistent with the challenges reported in foundational drift studies [*2*].
>
> *1. Wilson, Alphus D., and Manuela Baietto. "Applications and advances in electronic-nose technologies." sensors 9.7 (2009): 5099-5148.*
>
> *2. Vergara, Alexander, et al. "Chemical gas sensor drift compensation using classifier ensembles." Sensors and Actuators B: Chemical 166 (2012): 320-329.*

---

> ### Author Response · Authors · 2025-11-18
>
> **ii)** Noted. We have in fact already conducted experiments with several of the suggested methods. We initially omitted them from the paper due to mostly inferior performance and limited space. The results are summarized below:
>
> *Catboost*
> | Features    | Z00           | E11           | K29           | K76           | B18           | C34           | N18           | J44           |
> |-----------|---------------|---------------|---------------|---------------|---------------|---------------|---------------|---------------|
> | logfit_2  | 0.465 ± 0.012 | 0.516 ± 0.027 | 0.448 ± 0.057 | 0.612 ± 0.045 | 0.686 ± 0.033 | 0.503 ± 0.029 | 0.598 ± 0.040 | 0.448 ± 0.023 |
> | logfit_4  | 0.468 ± 0.006 | 0.478 ± 0.020 | 0.425 ± 0.033 | 0.600 ± 0.061 | 0.677 ± 0.028 | 0.503 ± 0.028 | 0.615 ± 0.028 | 0.452 ± 0.018 |
> | stats_3   | 0.507 ± 0.016 | 0.551 ± 0.028 | 0.453 ± 0.032 | 0.648 ± 0.027 | 0.698 ± 0.011 | 0.530 ± 0.012 | 0.541 ± 0.021 | 0.435 ± 0.025 |
> | stats_5   | 0.507 ± 0.016 | 0.551 ± 0.028 | 0.453 ± 0.032 | 0.648 ± 0.027 | 0.698 ± 0.011 | 0.530 ± 0.012 | 0.541 ± 0.021 | 0.435 ± 0.025 |
>
> *Xgboost*
> | Features    | Z00           | E11           | K29           | K76           | B18           | C34           | N18           | J44           |
> |-----------|---------------|---------------|---------------|---------------|---------------|---------------|---------------|---------------|
> | logfit_2  | 0.455 ± 0.040 | 0.508 ± 0.034 | 0.458 ± 0.028 | 0.545 ± 0.061 | 0.605 ± 0.057 | 0.549 ± 0.061 | 0.507 ± 0.025 | 0.462 ± 0.027 |
> | logfit_4  | 0.497 ± 0.008 | 0.501 ± 0.022 | 0.455 ± 0.049 | 0.546 ± 0.036 | 0.623 ± 0.063 | 0.541 ± 0.061 | 0.506 ± 0.040 | 0.491 ± 0.030 |
> | stats_3   | 0.565 ± 0.013 | 0.499 ± 0.044 | 0.548 ± 0.057 | 0.561 ± 0.057 | 0.607 ± 0.040 | 0.578 ± 0.035 | 0.528 ± 0.048 | 0.445 ± 0.046 |
> | stats_5   | 0.565 ± 0.013 | 0.499 ± 0.044 | 0.548 ± 0.057 | 0.561 ± 0.057 | 0.607 ± 0.040 | 0.578 ± 0.035 | 0.528 ± 0.048 | 0.445 ± 0.046 |
>
> *HistGradientBoost*
> | Features    | Z00           | E11           | K29           | K76           | B18           | C34           | N18           | J44           |
> |-----------|---------------|---------------|---------------|---------------|---------------|---------------|---------------|---------------|
> | logfit_2  | 0.411 ± 0.038 | 0.511 ± 0.045 | 0.476 ± 0.045 | 0.614 ± 0.042 | 0.610 ± 0.044 | 0.496 ± 0.036 | 0.521 ± 0.030 | 0.456 ± 0.037 |
> | logfit_4  | 0.440 ± 0.042 | 0.495 ± 0.031 | 0.470 ± 0.036 | 0.563 ± 0.043 | 0.676 ± 0.034 | 0.463 ± 0.032 | 0.515 ± 0.029 | 0.481 ± 0.033 |
> | stats_3   | 0.510 ± 0.029 | 0.526 ± 0.029 | 0.515 ± 0.060 | 0.615 ± 0.057 | 0.651 ± 0.030 | 0.580 ± 0.022 | 0.561 ± 0.045 | 0.429 ± 0.027 |
> | stats_5   | 0.510 ± 0.029 | 0.526 ± 0.029 | 0.515 ± 0.060 | 0.615 ± 0.057 | 0.651 ± 0.030 | 0.580 ± 0.022 | 0.561 ± 0.045 | 0.429 ± 0.027 |
>
> *RandomForest*
> | Features    | Z00           | E11           | K29           | K76           | B18           | C34           | N18           | J44           |
> |-----------|---------------|---------------|---------------|---------------|---------------|---------------|---------------|---------------|
> | logfit_2  | 0.479 ± 0.041 | 0.491 ± 0.044 | 0.471 ± 0.060 | 0.588 ± 0.045 | 0.627 ± 0.029 | 0.519 ± 0.043 | 0.530 ± 0.032 | 0.475 ± 0.043 |
> | logfit_4  | 0.482 ± 0.041 | 0.503 ± 0.041 | 0.466 ± 0.046 | 0.599 ± 0.051 | 0.630 ± 0.061 | 0.472 ± 0.047 | 0.547 ± 0.055 | 0.496 ± 0.031 |
> | stats_3   | 0.510 ± 0.032 | 0.477 ± 0.025 | 0.457 ± 0.051 | 0.556 ± 0.065 | 0.648 ± 0.035 | 0.505 ± 0.034 | 0.555 ± 0.044 | 0.420 ± 0.041 |
> | stats_5   | 0.504 ± 0.029 | 0.502 ± 0.048 | 0.448 ± 0.052 | 0.561 ± 0.080 | 0.648 ± 0.036 | 0.486 ± 0.027 | 0.534 ± 0.044 | 0.441 ± 0.035 |
>
> *ResNet18*
> | Features    | Z00           | E11           | K29           | K76           | B18           | C34           | N18           | J44           |
> |-----------|---------------|---------------|---------------|---------------|---------------|---------------|---------------|---------------|
> | polynomial  | 0.689↑ | 0.614↑ | 0.567↑ | 0.572↓ | 0.629↑ | 0.718↑ | 0.741↑ | 0.510↑ |
>
> In the revised paper, we will include the best-performing model from each key architecture class to provide a more comprehensive and fair benchmark while maintaining clarity in the presentation.

---

> > ### Comment · Reviewer_VA5c · 2025-11-26
> >
> > Thank you for your response. I appreciate the effort and results; I am changing my score, but some concerns (like multi-folds) are still not addressed.

---

> > > ### Author Response · Authors · 2025-11-26
> > >
> > > Thank you for your time and valuable comments! 🙏
> > >
> > > We agree that the temporal nature of the data poses a challenge for multi-fold validation. To address this, we have performed all experiments using random sampling of the training data to estimate the standard deviation of the reported AUC scores. We believe this addition will strengthen the statistical foundation of our results and help alleviate the remaining concerns.

---

### Author Response · Authors · 2025-11-26

Dear reviewers,

As we are incorporating the feedback and updating the paper, would you kindly assess/comment on our answers when you have a moment? We would be very grateful 🙏 if you could do so by the end of this week (*November 28, AoE*). This would allow us to include further enhancements.

---

### Author Response · Authors · 2025-12-02
**Revisions made and our final paper**

Dear Area Chair and Reviewers,

We were deeply disappointed by the OpenReview leakage incident and the subsequent disruption to the peer-review process. Like many authors and reviewers, we invested significant, honest effort into improving the paper based on the excellent feedback. We particularly noted that Reviewer VA5c acknowledged our efforts and revised their score upward from 4 to 6, while acknowledging that certain fundamental challenges inherent to longitudinal sensor data collection remain. Their final comment, "I appreciate the effort and results; I am changing my score," indicates that the substantive improvements were sufficient to warrant a higher score.

While we regret the inability to continue the discussion, we respectfully ask you to consider the substantial revisions made in our final paper and the corresponding author responses when making your decision.

Summary of key revisions made:
- We significantly broadened our experimental evaluation by adding eight new baseline methods, including ensemble methods (XGBoost, HGBoost, RF), fine-tuned ResNet18, and state-of-the-art time series models (InceptionTime, TS2Vec, LSTM, Graphormer). All results now report the mean and standard deviation of performance estimated via bootstrapping within the proposed temporal split, providing robust statistical validation within the constraints of our experimental design (Sec. 6.1, Table 4).

- We performed a dedicated analysis to disentangle demographic signals from the breath-based disease signal. We quantified the performance of models using only age and/or sex, and compared it to our full model's performance across demographic subgroups. This confirms that our models capture significant disease-related VOC signals beyond simple demographic correlations (Sec. 6.2, Table 5).

- We added a detailed comparison table situating the S-O-H dataset within the broader ecosystem of public olfactory resources, clearly establishing its unique value as the first large-scale, clinical breath time-series benchmark (Sec. 2, Table 1).

- We clarified all methodological details (e.g., sensor description, preprocessing, and data splits).

- In direct response to concerns about generalizability, we have already begun expanding the dataset into a multi-center resource. The initial 44 new tuberculosis samples from a second clinical site have been added to the public release, with more to follow.

The core contributions - a large, novel clinical dataset, a rigorous drift-aware benchmark, and an analysis of key deployment challenges - remain solid and are now presented with greater depth, clarity, and evidential support.

Thank you once again for the reviews 🙏. We believe the revised paper is significantly strengthened and provides a valuable, foundational resource for the machine learning and digital health communities. We are hopeful for a favorable decision.

---

### Meta-Review · Area_Chair_Zk5V · 2026-01-11

**Summary:**

All reviewers agree the dataset itself is the main contribution: a relatively large, clinically collected multivariate eNose time series dataset with a drift-aware temporal split that better reflects deployment than random splitting. Reviewers also consistently raised that the accompanying ML contribution is modest (standard CNN/CatBoost baselines), and that the empirical evidence, as originally presented, did not fully support several framing claims about robustness, generalization, and clinical screening utility. Key concerns centered on (1) incomplete and inconsistently described baselines, (2) limited statistical validation under a single (or very few) temporal split(s), (3) demographic and environmental confounding risks, (4) lack of explicit drift quantification or drift mitigation, and (5) limited evidence for cross-site or cross-cohort generalization.

**Reviewer Concerns:**

The rebuttal and revision improved the paper (expanded baselines, added variance estimates, added demographic-only and subgroup analyses, clarified typos/links, and added a dataset comparison table; plus initial steps toward multi-site expansion). However, the central gap remains: the submission is primarily a dataset/resource paper with limited methodological novelty and still limited evidence that reported performance reflects disease signal rather than confounds for key cohorts.

**Reviewer Scores:**

Discussion would likely converge to a mixed but slightly-positive set of scores, while still leaving the paper borderline under ICLR’s method novelty and evidence standards.

I would encourage the authors to continue dataset expansion (multi-site), add an evaluation protocol that supports multiple temporal folds (even if disease-specific folds differ), strengthen confound control (e.g., matched analyses or explicit adjustment), and report clinically meaningful operating points and calibration.

---

### Decision · Program_Chairs · 2026-01-26

Reject